# Health Access, Health Promotion, and Health Self-Management: Barriers When Building Comprehensive Ageing Communities

**DOI:** 10.3390/ijerph20196880

**Published:** 2023-10-03

**Authors:** Leticia Pérez-Saiz, Mireia Ferri Sanz, Maite Ferrando, Mirian Fernández Salido, Tamara Alhambra-Borrás, Jorge Garcés Ferrer, Rachael Dix

**Affiliations:** 1Kveloce I+D+I (Senior Europa S.L.), 46003 Valencia, Spain; mferri@kveloce.com (M.F.S.); mferrando@kveloce.com (M.F.); 2The Research Institute on Social Welfare Policy (POLIBIENESTAR), University of Valencia, Tarongers Campus, 46022 Valencia, Spain; mirian.fernandez-salido@uv.es (M.F.S.); tamara.alhambra@uv.es (T.A.-B.); jordi.garces@uv.es (J.G.F.); 3Center of Social and Urban Innovation Las Naves, 46024 Valencia, Spain; rachaeldix@gmail.com

**Keywords:** value-based care, healthy ageing, frailty, social care, barriers, health promotion, community, digital solution

## Abstract

A new intervention model for promoting healthy ageing grounded on integrated value-based care was developed and tested in the city of Valencia (Spain). Its implementation raised relevant barriers for older adults in their access to health, health promotion, and health self-management linked with their health and digital literacy. This new intervention model included several aspects. On the one hand, researchers together with older adults and their informal caregivers participating in the study, designed personalized care plans, based on older adults’ specific needs, to be implemented with the support of a digital solution. On the other hand, researchers and health and social professionals implemented a series of workshops in different locations of the city to encourage a sense of community among participants, reinforcing their trust in the new care model and increasing their adherence. Social activities were at the core of the workshops to understand older people’s interaction with the health and social services provided in the neighborhood. Qualitative and quantitative methods were combined to extract information from older participants on how to engage them as active actors of their health and understand their values and preferences. In the present manuscript, we focus on the qualitative results, which show that after a post-pandemic situation, they were more concerned about social isolation and desired face-to-face contact with their professional care team; however, feelings of loneliness and/or sadness were not considered among the reasons to visit health professionals. Some of the conclusions revealed that the use of technology as a supportive tool is well received but with a stress on its role as “supportive”, and not replacing the close contact with healthcare professionals. Professionals recognized the benefits of this new approach but required more time and incentives to dedicate the effort needed. The main aim of this study was to present these barriers related to health access, health promotion, and health self-management, as well as the actions developed to face them.

## 1. Introduction

According to the World Health Organisation (WHO) [1], healthy ageing is defined as the process of developing and maintaining the functional ability that enables wellbeing in older ages. Healthy ageing compiles a functional perspective on health in which personal factors, lifestyle behaviors, and environmental factors play crucial roles [2]. Nowadays, it entails a European societal challenge due to the current demographic change [3]. European healthcare systems are transforming the traditional model of care towards value-based care models, in which providers (healthcare and social professionals) together with patients and their families or informal caregivers aim to reach better health outcomes, reducing the incidence and effects of chronic diseases and living healthier lives [4]. Within this value-based care model, all the actors involved (healthcare and social professionals and informal caregivers) are engaged in a process of collaboration to define the specific needs of older people and to reach the best health outcomes. These value-based approaches enable older people to live the best life possible according to what they value, by facilitating access to high-quality, personalized, evidence-based, and integrated healthcare and social services.

In this context, ValueCare is a Horizon2020 EU project which aims at delivering efficient outcome-based integrated (health and social) care to older people facing cognitive impairment, frailty, and multiple chronic health conditions to improve their quality of life (and of their families) as well as the sustainability of the health and social care systems in Europe [5]. The ValueCare approach is supported by a digital solution, and it has been tested in seven large-scale pilots in Europe: Rijeka, Cork/Kerry, Coimbra, Treviso, Athens, Rotterdam, and Valencia [6].

In this framework, the authors of this paper implemented the intervention model for promoting healthy ageing in the city of Valencia (Spain) for frail older people. Frailty is a common clinical syndrome in older adults that increases their vulnerability because it entails poor health outcomes such as falls, disability, hospitalization, etc [7]. Although there is not a common and agreed definition of frailty [8], it is a common concern because of its cost implications in healthcare services [9]. The Spanish White Book on Frailty highlights that frailty is related to an increase in costs and financial burden on the health system. For that reason, this book promotes the implementation of cost-effective intervention programmes for frail older people [10]. Following this recommendation, the authors of this paper designed an integrated value-based programme for older adults with mild to moderate frailty to improve their care and quality of life by implementing value-based methodologies backed by a digital solution that integrates health and social care and offers a personalized care plan based on socio-health outcomes. The Valencia pilot aims to promote healthy lifestyles and reduce the burden of frailty in Valencia city, following the current healthcare reforms [11]. Indeed, value-based care is a current trend in healthcare management [12], where value is defined as the outcome that matter to patients about the related costs [13]. It is strongly linked with integrated care delivery as a systemic vision that responds to the needs of patients and focuses on health and wellbeing [14].

The proposed programme was based on four main pillars considering the advice of the clinical and social professionals working in the intervention programme: nutrition, physical activity, socialization, and adherence to medication. These four domains were highlighted as the main pillars to improve the wellbeing of frail older people in the city of Valencia. The four domains were aligned to change lifestyle behaviors and social environmental factors as two of the main domains of healthy ageing, as stated before. This was supported by a digital tool to facilitate the communication and monitoring of the programme between health professionals, researchers, informal caregivers, and participants. In this sense, technology has been recognized as a supportive tool for promoting healthy behaviors and alleviating the burden of traditional healthcare services [15]. 

However, some barriers were identified when implementing the programme in relation to health access, health promotion, and health self-management. These barriers have been identified and are presented in this paper, together with the means used to face them.

## 2. Detailed Case Description

### 2.1. Recruitment, Questionnaires, and Involvement in the Intervention Programme

The recruitment of frail older people in the city of Valencia (Spain) lasted eleven months. In the first step, participants were recruited from 7 different primary health care centres (all of them pertained to the same health department of the city of Valencia); and in the second step, older people were also recruited from 1 activity centre for older people, and 2 universities for older people. In the case of the health centres, health professionals (physicians and social workers) were contacted by the researchers and they, as knowledgeable about the medical history and personal lives of their patients, proposed the participation of those who could meet the requirements and be suitable for the proposed intervention programme. In the case of the other centres (universities and activity centres), coordinators were contacted, and the researchers carried out this first selection process based on an initial meeting with the voluntary older participants.

Once the first selection was performed, older people completed three questionnaires to measure their level of frailty (FRAIL scale [16]), level of dependency (Barthel [17]) and level of cognitive impairment (Pfeiffer SMPQ scale [18]) during a face-to-face interview with the healthcare professionals (in the case of older people recruited from the health centres) or with the project researchers (for those people recruited from the activity centre and universities). Thus, people over 60 years of age presenting mild to moderate levels of frailty without dependency or cognitive impairment were included in the intervention programme. Concretely, the inclusion criteria for participants were: (i) ≥ 60 years old; (ii) ≥ 1 on the FRAIL scale; (iii) ≥ 90 on the Barthel index; (iv) 0–2 on the Pfeiffer SMPQ scale; and (v) informed consent given. People who were not able to speak, read, or understand the Spanish language were not included in the intervention programme, as well as those who did not give informed consent. These criteria were agreed between researchers, doctors, and social workers involved in the study. A total of 120 older participants started the intervention.

Once recruited, the participants in a second face-to-face interview completed a baseline assessment based on the standardized questionnaire, particularly on the older adult version of the ICHOM dataset, including questions about quality of life, health and wellbeing, physical functioning, lifestyle, use of medication, received care, and sociodemographic information. The results of this questionnaire were used to determine the specific areas of intervention (nutrition, physical activity, use of medication, and/or socialization) in which participants needed to receive support and consequently, the modules to include in their corresponding care plans. 

The questionnaire results were explained by the researchers to each participant In a face-to-face meeting, which was the starting point for co-designing the intervention plans. Through a co-decision process among researchers and participants, with the support of health professionals (and in some cases, together with informal caregivers), the integrated care plans were designed. These care plans consisted of co-designed strategies where healthcare professionals could input requirements of health and care in a coordinated fashion for supporting older people in their specific needs, detected by all data collected with the baseline questionnaire. Within the care plans, personalized objectives were established within each area of intervention according to the specific needs of each participant. These objectives were weekly updated by the researchers as the intervention programme progressed, and participants could access and track their progress through a digital solution, which, apart from enabling the monitoring of the objectives, also allowed communication between health professionals/researchers and participants. The care plans were integrated into the digital solution to be prescribed to older people. Using a persuasive dialogue-based interface, participants were supported to accomplish the care plan since the digital solution can detect deviations and provide motivational feedback. Moreover, the digital solution enabled healthcare professionals to intervene when the goals were not achieved since they could visualize the participants’ performance in a specific dashboard.

### 2.2. Health Access, Health Promotion, and Health-Self Management Barriers

During the recruitment and the implementation of the baseline questionnaires, researchers spent a lot of time with the participants, seizing the opportunity not only to assess their health and wellbeing (quantitative) but also to better understand their health needs and concerns (qualitative). In this sense, researchers collected qualitative information that is presented in this paper on the main barriers detected by older people when accessing health and social services and their main concerns about having healthy lifestyles. 

#### 2.2.1. Feeling of Loneliness

Most of them expressed a feeling of loneliness. Some participants explained that the main reason for feeling unwanted loneliness was a consequence of the COVID-19 pandemic. They detected two main reasons: the first one was because they stopped attending the activities they used to take part in, and the second one was related to the increase in time spent at home alone. All these participants pointed out that to date, they had not recovered their pre-pandemic social life, which had affected their mental health and wellbeing. For other participants, the main cause of these feelings was the lack of contact with their relatives (especially children and grandchildren). In this sense, a high proportion of participants stated that they had not seen nor spoken to their relatives in the last few weeks, even in the last few months. 

However, all participants who reported feelings of loneliness did not consider them a reason to ask for help/support or to visit their healthcare professionals. In some specific cases, the healthcare professionals had become aware of these problems and had referred these people to the social worker, but most of these older people suffering from these feelings did not know about the existence of a social worker in their health centres, to whom they could ask for help. All this denotes a lack of communication within the health systems, among their professionals, and specifically between doctors and social workers working, sometimes, in the same building, and a lack of knowledge of this health service by potential users.

#### 2.2.2. Lack of Access to Healthcare (Regarding Our Areas of Intervention)

Another barrier encountered when implementing the value-based care approach was the existence of a general lack of information among the older population about the possibility to participate in activities related to three areas of the intervention (nutrition, physical activity, and/or socialization). For example, many participants were not aware of the activities related to physical or social activity that take place in their neighborhoods and which they can join. Regarding the area of nutrition, only older people with digestive tract diseases, diabetes, hypertension, or other related diseases had been provided with nutritional recommendations for a healthy diet. This lack of information leads authors back to the fact that the approach of healthcare systems for the older population is not focused on promoting healthy ageing and prevention but on the treatment of already developed diseases. 

#### 2.2.3. Difficulty in Handling the Digital Solution and Need for in-Person Contact

The use of the digital solution during the intervention was essential for communication between health professionals/researchers and participants. Researchers and professionals used the platform to set up care plans as well as to send specific objectives to the participants on a weekly basis. Participants used the app to complete the goals established by health and social care professionals and consult relevant information related to their goals. In addition, all of them (professionals, researchers, and participants) were able to track the progress in the digital tools. 

The use of the app through smartphones and tablets motivated participants to adhere to the care plan and follow a healthy lifestyle. Information about current and existing resources for healthy habits related to nutrition, physical activity, and socialization was incorporated into the supportive app linked with the intervention programme and into the objectives in an individualized way. However, the introduction to the new technology (app) was a challenge for a high proportion of older participants. They found it quite difficult and required the investment of time and effort to learn how to use it. Therefore, researchers designed dedicated digital training sessions to explain to older participants how to use the supportive app for the intervention programme. Despite the provision of training to each participant, all of them agreed on the high potential usability of the app, but as a support tool. They all stressed the importance and the need for closer contact, face-to-face, with the people in charge of managing their care plans. This is also related to the feeling of loneliness described above.

#### 2.2.4. More Time and Incentives Are Needed for Healthcare and Social Professionals to Dedicate the Required Effort

One of the barriers dealt with when implementing the intervention was the limited participation of health and social professionals, although they had a high interest in the value-based approach and in the study. Ideally, they were expected to be involved: (i) in the recruitment process, since they know their patients in-depth and know who presented a mild to moderate level of frailty; (ii) in the design of the care plans, to better adapt them to the specific needs of each participant; and (iii) in the follow-up of the intervention process. In general terms, however, that did not occur because time constraints also increased after the pandemic.

Both the design of the care plans and the establishment of weekly personalized objectives within three areas of intervention (nutrition, socialization, and physical activity) were based on the specific needs of each participant, from the results of the baseline questionnaire but also considering the received feedback during the face-to-face meetings. Moreover, regarding the objectives, they were developed also based on the participants’ personal goals, interests, capacities, and neighborhood resources, amongst other factors. Nevertheless, health professionals, both doctors and social workers, know their patients better than researchers and have the required knowledge that would have been very helpful in implementing an integrated value-based care approach for the establishment of the care plans and specific objectives, and throughout the intervention. However, only professionals from two out of the ten centres participating in the project were involved in some parts of the processes. All of them highlighted the lack of time to dedicate to extra tasks beyond their daily work, although the innovative approach of the project can, in the near future, alleviate their workload. All this points to the need for more time and incentives for healthcare professionals to dedicate their efforts to implement the value-based care approach. 

### 2.3. Activities to Tackle the Mentioned Barriers while Promoting Healthy Ageing

Researchers implemented some extra activities to promote the participation of older people in the study and increase their adherence to the intervention proposed in three of the areas (nutrition, physical activity, and socialization). 

#### 2.3.1. Training Sessions in Digital Solution

As anticipated above, due to the important role of the app for the participants as well as the difficulty in its management, researchers conducted a series of training sessions to teach the intervention group the different functions of this part of the digital solution and how to use them. These training sessions had a very positive effect on the participants, not only in terms of learning about new technologies, but also in terms of feeling involved in controlling their health and wellbeing. Making decisions about their goals, handling an application that was completely new to them, and finally achieving these specific objectives resulted in the empowerment of older people participating in the project.

In addition, training sessions on the platform were also given to health professionals and social workers in the centres participating in the study. The main objectives of these sessions were to demonstrate the important role that these professionals could have in the intervention of each of their patients, as well as to teach them how to use all the tools available on the platform and make them aware of the benefits of using the app for older people.

#### 2.3.2. Workshops

The difficulty for participants in using the app, together with their manifested feelings of loneliness and/or sadness and their preference/need for in-person contact were the reasons why researchers included a series of workshops in the intervention process. Moreover, the overall aim of the intervention was to increase the quality of life of the defined target group, avoid loneliness, and provide tools for healthy lifestyles. Thus, the development of these motivational group sessions emerged as an opportunity for contributing to tackling the barriers described above while fostering healthy ageing amongst participants.

These workshops were prescribed to the participants in the intervention group as part of their personalized care plan and were focused on different topics, according to the findings discovered in the codesign based on the interests and needs of the older people;Iey also considered the important themes related to healthy ageing from the relevant literature [19]. The topics addressed in the workshops were: Healthy Eating for Unwanted Loneliness; Art and Cognitive Stimulation; Ageism and Older Person Abuse; Managing Emotions; Technology Tools; and finally, an open-air multi-exercise class. They were performed over six months, from December 2022 to May 2023, delivering one topic each month. The strategy was to offer these activities to the participants in their own healthcare centre/activity centre/university, so ideally, there should have been a workshop for each topic in each of the participating centres. However, as these centres were quite dispersed and due to limited resources, a minimum of three and a maximum of six workshops were performed each month, trying to group the patients from the nearby centres. This number was also adapted to the availability of external experts who delivered some of the workshops. 

Thus, the implementation of these sessions tackled the unwanted loneliness while contributing to improving behavioral habits regarding nutrition and physical activity. Therefore, attendance at workshops aroused the older people’s interest in learning more about the subjects in which these events were focused. 

It is important to note that healthcare professionals were also invited to participate in these group sessions to foster their motivation for being involved in the intervention.

#### 2.3.3. Motivational Interviews

Together with the establishment of the objectives through the digital solution and the implementation of workshops, motivational interviews were carried out as part of the intervention. The use of motivational interviewing was specifically focused on addressing psychosocial fragility with special attention to those who have experienced feelings of loneliness. The motivational interviewing allowed a collaborative and person-centred approach to create a relationship of profes—onal–patient trust, in which the professional’s attitude was one of acceptance and empathy towards the patient’s needs, preferences, and experiences. In addition, from this approach, the application of motivation increased the patient’s intrinsic personal motivation and commitment to change towards healthier lifestyles, with the support of professionals to explore and resolve the ambivalences that hindered the initiation and maintenance of these healthier behavioral habits.

## 3. Discussion

In line with the urgent need to promote healthy ageing focused on value-based integrated care models, the authors implemented a community-integrated model in which elderly people, their health professionals and informal caregivers were involved. On the one hand, a specific care plan was designed for participants, adapted to their needs and values; and on the other hand, workshops focused on different topics were carried out to foster a sense of community amongst older people as well as their confidence in this new model. In addition, motivational interviews and training sessions were implemented. However, during the intervention, important barriers were encountered in the implementation of this value-based care model.

Firstly, to develop a sense of community, older people need to generate and be involved in this community. However, there is a general lack of information about the activities that take place in their neighborhoods and the possibility of attending. This lack of access to healthcare is directly related to a lack of coordination between doctors and social workers; when older people are sick, they visit their doctor and receive treatment, usually pharmacological. However, when they need to be referred to the social worker at the health centre, information about the patient does not flow as it should among these health professionals, resulting in the referred lack of coordination, and consequently, in a health system which is not focused on promoting healthy ageing and preventing diseases among older people. Currently, there is a growing awareness of fragmentation of care leading both to inefficiency and adverse outcomes in multimorbid older patients [20]. A recent study observed ‘an absence of a shared and complete understanding of frailty amongst healthcare professionals’ and ‘a fragmented model of care for community-dwelling frail older patients’ [21]. Thus, coordination of care is still one of the challenges of treating older adults [21], and addressing this issue is crucial to move forward towards a model of care based on the community.

This lack of coordination is often due to the shortage of time and the excessive workload that characterizes health professionals. This was another of the barriers that authors faced during the implementation of the intervention, which led to a lack of professionals’ involvement in the development and follow-up of the care plans. Healthcare workers experience challenging situations of extreme stress and pressure, which leads to work overload from working overtime [22]. This lack of time and high workload, in turn, leads to the improper delivery of patient care and affects the quality of job performance of the healthcare systems [23,24]. While there is no simple solution to decrease the impact of burnout, specific strategies are needed to keep a healthy workplaces framework and maintain a high quality of care [25]. 

The use of new technologies could be a key factor in both lessening the burden on healthcare systems and empowering older people to become increasingly involved in the control of their health and wellbeing. In the last few years, digitalization has become a necessity for ensuring older adults’ need for information, services, and social inclusion [26]. Due to the increasing digitalization of society, and particularly because of the impact of the COVID-19 pandemic on our lives, older people need to adapt to this situation by acquiring digital competencies [27]. In the study, participants used the digital solution (the app developed), to follow the care plan designed specifically for them. From this app, participants were able to monitor their goals, while improving the communication with their formal caregivers to receive advice and feedback. In this way, professionals could reduce their workload by following up with some patients online, while the latter acquired a much more active role in the control and management of their health and wellbeing. However, many older people found it difficult to use the digital solution. The use of new technologies has widely been reported as a particular challenge for the elderly since they have limited abilities to access and use the Internet, and, therefore, miss out on the benefits of online services [28,29]. Although current and emerging technologies play an important role both in reducing the impact of many of the daily challenges older adults face and in facilitating the addressing of their individual needs [30], individuals usually find barriers in using them, mostly due to the ignorance of technological features [31]. 

In addition, during the intervention programme, participants reported a feeling of unwanted loneliness. This was the reason, together with the difficulty in using the app, to focus the intervention on conducting a series of workshops addressed to foster healthier habits, allowing the older people to meet with other individuals from the community, and establishing the digital solution as a support tool. Older people are especially vulnerable to loneliness and social isolation, particularly after a pandemic situation. Loneliness is a serious yet underappreciated public health risk that affects a significant portion of the older adult population. The relationships between loneliness and health impacts can be reciprocal in that not only can feeling lonely have an impact on health, but the resultant health conditions can increase an individual’s likelihood of experiencing loneliness [32]. However, older people do not consider this feeling of loneliness a reason to visit the doctor. Referring to a value-based care system, this feeling of unwanted loneliness should be addressed from the perspective of the community, always considering the specific needs of everyone. This is exactly what the authors intended to do during the workshops’ implementation.

## 4. Conclusions

There is an urgent need for healthcare systems’ transformation towards value-based models of care, based on personalized solutions, according to the specific needs and requirements of patients. This transformation is a challenge for both professionals and older people. The latter play a crucial role, becoming fully involved and acquiring an active role in the management of their health and wellbeing, aiming at the healthiest possible ageing. In this sense and considering the barriers that appeared at the first stage of the intervention, the authors adjusted the intervention to the specific needs of older people. The adaptations made were: (i) a specific care plan for each participant, addressing their needs about physical activity, nutrition, and socialization; (ii) a digital solution (and the required training) was offered as a support tool to follow the care plan process and to improve the communication between patients and health professionals; (iii) implementation of different face-to-face workshops and motivational interviews to reduce the feeling of unwanted loneliness. All these activities had the ambition of fostering the adoption of a healthier lifestyle leading to healthier ageing, where older people play a crucial role in the management of their health and wellbeing with the support of both health professionals and informal caregivers. Healthcare professionals support this shift to a value-based system of care. Nevertheless, although they agree to put all their efforts into these community-integrated models, they do not have the necessary time due to high workloads. Therefore, community-integrated models based on value are well received by professionals and older people, but a paradigm shift is needed. In this framework, technology can be used as a powerful tool to support the provision of care. Proper use of these technological tools can facilitate communication between patients and professionals, as well as help older people keep track of and monitor their health conditions and their evolution. However, during the implementation of the project, loneliness was one of the main concerns highlighted by older people, so the use of these new technologies should not replace face-to-face contact with social and health professionals but be used as support tools for the empowerment of people to take an active role in the control of their healtI.

The implementation of our intervention will entail signIficant implications, not only at the scientific level but also at the technological and societal levels. Firstly, our study will provide evidence-based results on the barriers of older people regarding health access, health promotion, and health self-management, supporting the strong need for a switch from traditional models of care towards those which are based on value. Moreover, the intervention will have an impact on technology and IT tools in value-based integrated care models, since we will provide evidence-based results of how the management of the digital solution empowers older people to self-control their health. Finally, by integrating all the required information in a personalized care plan and including it in the digital solution, the efficiency of care will be improved, and consequently, the satisfaction of patients and the motivation of healthcare professionals.

## Data Availability

Data sharing is not applicable.

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
