# Peer review of "Health Access, Health Promotion, and Health Self-Management: Barriers When Building Comprehensive Ageing Communities"

_ijerph, 2023, doi:10.3390/ijerph20196880_

Round 1

Reviewer 1 Report

Thank you very much for allowing me to review this manuscript. The subject matter is very interesting, we are facing a problem in society such as frailty, vulnerability and loneliness in the elderly. This manuscript needs a number of improvements.

In the introduction the authors refer to the design of a model developed in this study to promote lifestyles and reduce frailty in older people. However, they make no reference to the basis of this model or the theories underpinning it (value-based or needs-based). Perhaps this part should be in the methodology section and the introduction should be based on studies or concepts that refer to this model, including initiatives or research in other countries that support the intervention carried out.

The recruitment of frail older people in the city of Valencia (Spain) lasted more than 70 eight months. How long was this exactly?

How did you contact the older people and what was the recruitment like?

Specify what the questionnaires used measure.

What do the care plans of older people refer to? Why were no nurses involved in the study when referring to promotion and care plans? More information is needed about these care plans.

The authors should detail how this quantitative and/or qualitative research was done. From the results they show it seems to have been more qualitative, how was it done?

What limitations did they find in this study, and is future research planned?

What implications does this study have for health and social systems?

Reviewer 2 Report

The introduction part presents a value-based program insufficiently clearly. It is advised to provide a more thorough explanation as the definition of value is difficult to comprehend.

Quantitative results are misrepresented in the Detailed Case Description section. To improve scientific research and data support, it is advised to provide precise data results after using quantitative approaches, such as the result analysis following a questionnaire survey.

1.I don't see a clear indication of the primary issue that your study is attempting to solve in your abstract section, and I would recommend adding information about this issue to the abstract.

2.Your abstract indicates that the work employs a mix of quantitative and qualitative methodologies, however I don't see any data graphs connected to quantitative research throughout the article. I recommend adding graphics to the article to represent your quantitative data.

3.You indicated that you employed a questionnaire in the section titled "Health access, health promotion and health-self management barriers" and that you might put the findings of your survey in graphical form in the margins so that readers could come and better comprehend your content.

Round 2

Reviewer 1 Report

The authors made all the suggested contributions, therefore, it can be accepted.

Author Response

In this second round, the only reviewer comment is:

"The authors made all the suggested contributions, therefore, it can be accepted"